# Use of Validated Questionnaires to Predict Cosmetic Outcomes of Hypospadias Repair

**DOI:** 10.3390/children11020189

**Published:** 2024-02-02

**Authors:** Amos Neheman, Omri Schwarztuch Gildor, Andrew Shumaker, Ilia Beberashvili, Yuval Bar-Yosef, Shmuel Arnon, Amnon Zisman, Kobi Stav

**Affiliations:** 1Meir Medical Center, Department of Urology, Kfar Saba 44281, Israel; amosne@clalit.org.il (A.N.);; 2Faculty of Medicine, Tel Aviv University, Tel Aviv 6997801, Israel; shumaka4@ccf.org.il (A.S.); iliab@asaf.health.gov.il (I.B.); yuvalby@tlvmc.gov.il (Y.B.-Y.); amnonz@asaf.health.gov.il (A.Z.); stavkobi@gmail.com (K.S.); 3Shamir Medical Center, Department of Urology, Zerifin 703001, Israel; 4Shamir Medical Center, Department of Nephrology, Zerifin 703001, Israel; 5Department of Pediatric Urology, Dana-Dwek Children’s Hospital, Tel Aviv Medical Center, Tel Aviv 6423906, Israel; 6Meir Medical Center, Department of Neonatology, Kfar Saba 4428164, Israel

**Keywords:** hypospadias, HOPE score, GMS score, cosmetic results, tubularized incised plate urethroplasty

## Abstract

Introduction: Hypospadias is a syndrome of penile maldevelopment. The primary goal of hypospadias surgery is to create a penis with normal appearance and function. Historically, the outcome of hypospadias repair has been assessed based on the need for reoperation due to urethroplasty complications (UC), including fistula formation, dehiscence, meatal stenosis, or development of a urethral stricture. The Glans–Urethral Meatus–Shaft (GMS) score is a standardized tool to predict UC. Analysis of the cosmetic outcomes of hypospadias repair based on the appearance of the reconstructed penis has been validated, and standardized scores have been published. The Hypospadias Objective Penile Evaluation (HOPE) score is a validated questionnaire used to assess postoperative cosmetic outcomes. Although predictors of surgical outcomes and UC have been well documented, predictors of optimal cosmetic outcomes are lacking in the literature. Furthermore, reoperation due to cosmetic considerations has been poorly reported. Objective: To identify predictors of cosmetic outcomes after hypospadias repair and to assess the reoperation rate according to cosmetic considerations. Materials and Methods: This prospective cohort study included 126 boys who underwent primary hypospadias repair. The severity of hypospadias, degree of penile curvature, glans width, preoperative HOPE, and GMS scores were documented. The standard technique for single-stage repairs, the tubularized incised plate urethroplasty, was performed. The primary endpoint was cosmetic outcomes evaluated by the HOPE score questionnaire six months postoperatively. Optimal cosmetic results were defined by HOPE scores ≥ 57. Results: The study population consisted of the following cases: 87 (69%) subcoronal, 32 (25%) shaft, and 7 (6%) proximal hypospadias. Among the study participants, 102 boys (81%) had optimal cosmetic results (HOPE ≥ 57), and 24 boys (19%) had surgeries with suboptimal cosmetic outcomes (HOPE < 57). Ancillary procedures were performed in 21 boys (16%), of which 14 (11%) were solely for cosmetic considerations, and 7 were secondary to UC. Using the Receiver Operating Characteristic analysis of potential predictors of optimal cosmetic outcomes, the preoperative HOPE score had the highest area under the curve (AUC = 0.79; 95% CI 0.69–0.89, *p* < 0.001). After multivariable analysis, the degree of penile chordee (*p* = 0.013), glans width (*p* = 0.003), GMS score (*p* = 0.007), and preoperative HOPE score (*p* = 0.002) were significant predictors of cosmetic outcomes. Although meatal location predicted suboptimal cosmetic results in univariate analysis, it was not a factor in multivariable analysis. Conclusions: Over 80% of boys undergoing hypospadias repair achieved optimal cosmetic outcomes. More than 10% of cases underwent ancillary procedures, secondary solely to cosmetic considerations. Predictors of optimal cosmetic outcomes after hypospadias surgery included degree of chordee, glans width, and preoperative HOPE and GMS scores, which were the best predictors of satisfactory cosmetic results. Although meatal location is the main predictor of UC, it was not a predictor for cosmetic outcomes. Factors affecting cosmetic outcomes should be clearly explained to parents during the preoperative consultation.

## 1. Introduction

Hypospadias is a syndrome of penile maldevelopment that occurs in 1:200 males [1,2]. The primary goal of hypospadias surgery is to create a penis with normal appearance and function.

Historically, the outcome of hypospadias repair has been assessed based on the need for reoperation due to urethroplasty complications (UC). According to a meta-analysis of 49 studies with 4675 boys, the overall complication rate was 10.6%, with a reoperation rate of 4.5%. Typical reasons included fistula formation (5.7%), meatal stenosis (3.6%), and development of urethral stricture (1.3%) [3]. Dehiscence is another potential complication in hypospadias surgery, occurring in 2–15% of cases [4,5]. While the primary reason for reoperation is UC, cosmetic concerns account for up to 31.3% of reoperations [6].

The Postoperative Hypospadias Objective Penile Evaluation (HOPE) score [7] is used to evaluate cosmetic outcomes based on the configuration of the glans, appearance of the meatus, penile skin properties, and residual curvature. Analysis of the cosmetic outcomes of hypospadias repair based on the appearance of the reconstructed penis has been reported, and standardized scores have been published. This study aimed to identify preoperative predictors of cosmetic outcomes and assess the reoperation rate based on cosmetic considerations according to the GMS and HOPE scores. We hypothesized that the independent predictors used to define potential complications of hypospadias repair would also predict cosmetic outcomes.

## 2. Materials and Methods

### 2.1. Study Population

This prospective cohort study included 154 boys who underwent primary hypospadias repair from January 2016 through February 2020. All cases of hypospadias were consecutive and performed by a single surgeon.

The inclusion criteria were boys with congenital hypospadias, otherwise healthy, and planned for one-stage hypospadias repair. Exclusion criteria were scrotal or perineal hypospadias, treatment with preoperative testosterone, treatment with hyperbaric oxygen, patients who underwent a planned staged hypospadias repair, and those who were lost to follow-up.

Twenty-two boys were excluded based on these criteria. Among the remaining 132 boys, 6 were lost to follow-up and 126 were included in the final analysis (Figure 1).

### 2.2. Study Tools

The Hypospadias Objective Penile Evaluation (HOPE) score [7] includes six items, each receiving a score of 1 to 10 according to a set of pictures serving as the index value. Each item evaluates a different component: the position of the meatus, the shape of the meatus, the shape of the glans, the shape of the penile skin, the penile axis, and penile curvature (Figure 2). The maximum HOPE score is 60.

We defined strict criteria for an optimal HOPE score of ≥57. A HOPE score < 57 was considered a suboptimal cosmetic outcome.

Preoperative risk factors for developing urethroplasty complications have been evaluated using the Glans–Urethral Meatus–Shaft (GMS) score [8,9]. The GMS score assesses the severity of hypospadias based on the size of the glans, quality of the urethral plate, meatal position, and degree of curvature. Each of these three components is scored numerically on a scale of 1 to 4, with more unfavorable characteristics being assigned higher values. The values are then summed to determine the GMS score. The lowest possible GMS score is 3 (very mild hypospadias), and the highest score is 12 (severe hypospadias) (Figure 3). The score is classified into three levels: mild (3–6), moderate (7–9), and severe (10–12). There is a correlation between the GMS score and postoperative UC [10]. However, additional studies examining predictors of poor cosmetic outcomes in repeat operations are needed.

### 2.3. Pre-Urethroplasty Information

Demographic information, including birth weight and gestational age at birth, as well as clinical and physical examination parameters, were obtained during the preoperative assessment. Meatal location was stratified to distal (subcoronal), shaft, or proximal (penoscrotal). Preoperative HOPE [7] and GMS scores for each case were evaluated by the surgeon.

At the beginning of the operation, at the time of examination under anesthesia, the preoperative width of the glans was measured using a caliper, and the degree of curvature (after degloving and performing an erection test) was assessed using a goniometer. The severity of curvature was classified as mild (0–30°), moderate (30–60°), or severe (>60°). Correction of chordee is dictated by its severity, as described in a meta-analysis by Babu et al. [11]. In the 17 articles included in the meta-analysis, the severity of chordee was evaluated after degloving [12]. The surgical technique was chosen after degloving and releasing tethering bands [13].

### 2.4. Surgical Technique

Tubularized incised plate urethroplasty was the preferred operative technique in the present study [14]. A longitudinal relaxing incision in the urethral plate was performed. Urethroplasty was performed using two layers of 7/0 polydioxanone subcuticular running sutures. An additional layer of dartos fascia overlying the urethroplasty was harvested from the ventral aspect of the penis. Urethroplasty was performed over an 8F stent. We used an 8F Zaontz stent (Cook Medical, Bloomington, IN, USA) for all patients [15].

The penile straightening procedure was dictated by the severity of curvature. Mild curvature (<30°) was managed using dorsal tunica albuginea plications. Moderate curvature (30–60^o^) was treated by mobilizing the neurovascular bundle and using a dorsal Heineke–Mikulicz incision with nonabsorbable sutures. Severe curvature (>60°) was treated with three deep ventral corporotomies reinforced with dorsal plication [11].

### 2.5. Study Outcome Measures

The primary endpoint was the HOPE score assessed six months postoperatively by the surgeon. In cases when an ancillary procedure was performed secondary to UC or cosmetic considerations, the HOPE score was assessed six months after the most recent procedure. Ancillary procedures were considered when parents sought better cosmetic outcomes based on the parameters of redundant foreskin, penile torsion, presence of inclusion cysts, or residual curvature, along with the surgeon’s assessment that they were achievable through surgery. Urethroplasty complications were not included in the study endpoints; however, they were included in the statistical analysis.

The study cohort was divided into two groups according to postoperative HOPE scores that were subsequently compared. These groups were boys with optimal cosmetic outcomes (HOPE ≥ 57) and boys with suboptimal cosmetic outcomes (HOPE < 57).

### 2.6. Statistics

#### Sample Size Calculation

The sample size was calculated using G∗Power software, version 3.1.9.7. We planned to include 136 patients. The calculated sample size (at least 22 patients in each group) allowed us to have an 80% chance, with a two-sided significance of 0.05, to detect a difference of 3 points in the preoperative HOPE score between boys with optimal (≥57) and suboptimal (<57) HOPE scores. A difference of 3 points in the preoperative HOPE score may be clinically significant in predicting optimal cosmetic outcomes. In the sample size calculation, we also considered an incidence of undesirable cosmetic results from the surgery of about 20% and a potential dropout rate of the study participants of approximately 15%.

Statistical analyses were performed using SPSS software, version 28.0 (IBM Corp., Armonk, NY, USA). A *p*-value < 0.05 was considered significant for all tests. Normally distributed data were expressed as mean ± SD. Medians and interquartile ranges (quartiles 1–3) were used for variables that did not follow a normal distribution. Categorical variables were described as frequencies. The Kolmogorov–Smirnov test was conducted to determine whether the study variables were normally distributed. Normally distributed continuous variables were compared between the two groups using t-tests. Variables with skewed distribution were compared using nonparametric Mann–Whitney U tests. The chi-square test was used for comparison of categorical variables. The areas under the Receiver Operating Characteristic (ROC) curves were computed to describe the discrimination ability of the potential predictors of the HOPE score. Univariate and multivariate logistic regression analyses were performed to determine the predictors of the HOPE score. The parameters from univariate data analyses with *p* < 0.25 for predicting outcomes were selected as confounders for multivariable models. Univariate and multivariate logistic regression analyses are presented as odds ratios (OR) with 95% confidence intervals (CI). Some parameters were analyzed as numerical and nominal (dichotomous) variables.

## 3. Results

The study population consisted of 126 boys. There were 87 cases of subcoronal (69%), 32 of shaft (25%), and 7 with proximal hypospadias (6%). Penile curvature was classified as mild/none in 73 boys (58%), moderate in 42 boys (33%), and severe in 11 boys (9%).

The group with optimal cosmetic results (HOPE ≥ 57) consisted of 102 boys (81%), and the group with suboptimal cosmetic outcomes (HOPE < 57) included 24 boys (19%). There were no significant differences between the groups regarding demographics. However, there were significant differences regarding hypospadias severity, glans width, preoperative HOPE score, and preoperative GMS score (Table 1). In the optimal cosmetic results group, 73% of the boys had subcoronal hypospadias vs. only 54% in the suboptimal cosmetic results group (*p* = 0.013). Furthermore, the degree of the chordee was classified as mild, moderate, and severe in 67%, 25%, and 6% of the boys in the optimal cosmetics results group vs. 20%, 60%, and 20% in the suboptimal cosmetics results group, respectively (*p* < 0.001). The GSM score was lower in the optimal cosmetics results group, and the preoperative HOPE score was higher in this group (*p* < 0.001).

During a mean follow-up of 11 months, the following urethroplasty complications occurred: urethral fistula developed in 10 (8%), glans dehiscence in 5 (4%), and meatal stenosis in 7 boys (5.5%).

Ancillary procedures were performed in 21 boys (16%), 14 solely for cosmetic considerations (11%) and 7 secondary to UC (5%). Procedures performed included revision of circumcision in 11 cases, penile straightening due to residual chordee in 8 cases, and excision of an inclusion cyst in 2 cases.

ROC curve analyses were performed to determine potential predictors with discriminating ability for suboptimal cosmetic outcomes (Table 2 and Figure 4).

The preoperative HOPE score demonstrated the largest area under the ROC curve (AUC) (0.79, 95% CI 0.69–0.89, *p* < 0.001) for predicting suboptimal cosmetic outcomes, with a sensitivity and specificity of 76% under a cutoff of 45. The GMS score and degree of curvature were also significant predictors; however, not as strongly as the preoperative HOPE score.

The degree of curvature, glans width, GMS score, and preoperative HOPE score was defined as dichotomization, and were significant predictors of cosmetic outcomes in univariate logistic regression analysis. In this respect, the preoperative HOPE score was the strongest predictor; a value ≥45 predicted 85% odds of achieving optimal cosmetic outcomes. Although meatal location (categorized as subcoronal, shaft, and proximal) served as a predictor of suboptimal cosmetic results in univariate analysis, it did not follow multivariable adjustments (Table 3).

## 4. Discussion

Hypospadias parameters evaluated prior to surgery affect cosmetic surgical outcomes. The degree of curvature, glans width, GMS score, and preoperative HOPE score, but not meatal location, were predictors of cosmetic results.

Our hypothesis that predictors of cosmetic outcomes will reflect those predicting UC was only partially confirmed by the study results. Meatal location defining the severity of hypospadias was the primary known predictor of UC but not of cosmetic outcomes [8].

Achieving satisfactory cosmetic outcomes propels the endless evolution of modern hypospadias repair techniques [16]. Nevertheless, there are no guidelines to assist in assessing surgical outcomes. While UCs are routinely and uniformly documented and addressed accordingly, cosmetic outcomes are usually assessed during a research study [3,17]. Furthermore, there is a lack of consensus on how to address suboptimal outcomes. The definition of “good cosmetic outcomes” is not well described. Baskin suggested that the standard should be to create a penis that would be considered a cosmetically normal, circumcised penis [18].

Several tools for evaluating postoperative cosmetic outcomes were published [17,19,20]. We used the HOPE score because it is a validated questionnaire, relatively objective, and easy to use in a clinical setting [7]. However, the HOPE score is assessed by the surgeon, and it has been described that physicians reported better cosmetic outcomes when compared to parents [21,22].

In the present study, the distribution of meatal location is similar to that reported in larger series, where approximately 70–85% were distal [23]. UCs were not an endpoint in this study but were documented. Urethral–cutaneous fistula developed in 8%, dehiscence in 4%, and meatal stenosis in 5.5%, which was about 17% of the study cohort overall. This is similar to other reported series on hypospadias [24]. Studies have reported a range of fistula rates after hypospadias repair, which can be influenced by factors such as the severity of hypospadias, the age of the patient at the time of surgery, and the presence of associated conditions. Reported rates vary from 5% to 30–40% [25,26].

To the best of our knowledge, this study is the first attempt to focus on and determine independent predictors of cosmetic outcomes, regardless of UC.

In this study, more than 80% of surgeries resulted in satisfactory cosmetic outcomes. In a review article, Van der Horst and de Wall stated that the goal of hypospadias repair is to achieve cosmetic and functional normality [27]. In their article, good cosmetic outcomes were achieved in 70% of cases based on self-reported questionnaires of adults who underwent hypospadias surgery in childhood [28]. In the group of patients with proximal hypospadias, only 50% reported satisfactory outcomes. It is noteworthy that the results in the present study, based on validated questionnaires and evaluated by a surgeon, were similar to those of adults who underwent surgery in childhood and reported their satisfaction with their penile appearance. The DRAQULA study [29] assessed decision regret and quality of life assessment in adolescents who underwent hypospadias repair in childhood. They reported that 90% of the patients were satisfied with early hypospadias surgery, with average health-related quality of life scores and low levels of decisional regret among patients and parents.

There is a paucity of information in the literature regarding ancillary procedures secondary to cosmetic considerations. In our study, 14 boys (11%) underwent an ancillary procedure solely for cosmetic considerations, most for revision of circumcision and residual chordee repair. Spinoit et al. retrospectively evaluated 543 patients operated in a tertiary center. The reoperation rate due to cosmetic outcomes was 31%. This is higher than what we found in our study. In their article, they did not specify the reason for surgery or the cases that were operated on solely because of cosmetic reasons combined with UC. This may explain the higher rate of ancillary procedures performed in that study [30].

The preoperative GMS score is a brief and exact method with good inter-observer reliability for describing the severity of hypospadias [9]. Additionally, it was previously found to correlate with UC-related surgical outcomes. Interestingly, we found that the GMS score is a predictor of cosmetic outcomes as well.

The position of the meatus as a classification of the severity of hypospadias is an established predictor of UC [6]. However, after multivariate adjustment, it did not remain an independent predictor of suboptimal outcomes leading to reoperation for cosmetic reasons. Conversely, the severity of curvature and glans properties were found to be predictors of cosmetic outcomes. This finding is counterintuitive to what we may think of as predictors of successful hypospadias surgery, which may indicate that we should address functional and cosmetic outcome prognosticators separately rather than as a single entity.

The HOPE score was validated previously as a tool to evaluate postoperative cosmetic outcomes [7]. To the best of our knowledge, the present study is the first attempt to utilize the HOPE score as a preoperative prognostic tool. A HOPE score below the cutoff value of 45 appeared to be the strongest predictor of poor cosmetic outcomes, with the largest AUC.

## 5. Limitations

This study had several limitations. It presents the experience of a single surgeon at one academic center. Therefore, the results may not be generalizable and should be confirmed by larger, multi-center studies. However, surgery performed by a single surgeon removes the bias associated with variations in technique and decision-making processes associated with cohorts involving several surgeons. Outcomes were also evaluated by the same single surgeon. This is because child–parent–doctor relationships in our research setting are monitored and well controlled. All boys and parents meet the same nurse practitioner and treating physician. Other personnel do not enter when the genitalia are being examined [31]. In the HOPE score validation study, the intra-observer reliability had an average Pearson correlation coefficient of 0.817, reflecting strong intra-rater reproducibility. The inter-observer reliability demonstrated an intraclass correlation coefficient of 0.790, indicating sufficient agreement among observers. These results imply that additional evaluations are unlikely to change the results dramatically.

Another limitation of the study was that the HOPE score has not yet been validated for preoperative evaluation. While meatal location and the degree of curvature (if measured uniformly with a caliper) show low inter-observer variation, other factors in the HOPE score, such as the appearance of the glans and meatus, can exhibit higher rates of inter-observer variability and should be validated. Additionally, using the HOPE score for both preoperative assessment and as an outcome measure has the potential to introduce bias. Given the current absence of validated tools to assess preoperative cosmetic appearance, we used the HOPE score for this purpose, as in other surgical procedures where cosmetic outcomes are crucial. Patniak et al. employed the Derriford Appearance Scale to evaluate cosmetic outcomes in rhinoplasty; the same scale was used both preoperatively and three months postoperatively [32]. Rosa et al. validated the Utrecht questionnaire for outcome assessment in aesthetic rhinoplasty, using the same questionnaire pre- and postoperatively [33]. Ghilli et al. reported on the quality of life and cosmetic satisfaction using the BREAST-Q questionnaire before and after oncoplastic or traditional breast-conserving surgery [34]. Based on the study design, we cannot draw any conclusions regarding parental perceptions, patient satisfaction, psychosocial aspects, or future sexual function [28]. Indications for ancillary intervention due to cosmetic considerations were not clearly defined and relied on surgeon recommendations and parental preferences. Defining strict cosmetic criteria for performing ancillary procedures could dramatically influence the prevalence (decrease or increase) of procedures performed.

## 6. Conclusions

Over 80% of boys undergoing hypospadias repair achieve optimal cosmetic outcomes. More than 10% of cases undergo ancillary procedures secondary solely to cosmetic considerations. Predictors of optimal cosmetic outcomes after hypospadias surgery include degree of chordee, glans width, and preoperative HOPE and GMS scores, which are the best predictors of satisfactory cosmetic results. Although meatal location is the main predictor for postoperative complications, it is not a predictor of cosmetic outcomes. Factors affecting cosmetic outcomes should be clearly explained to parents during the preoperative consultation.

## Figures and Tables

**Figure 1 children-11-00189-f001:**
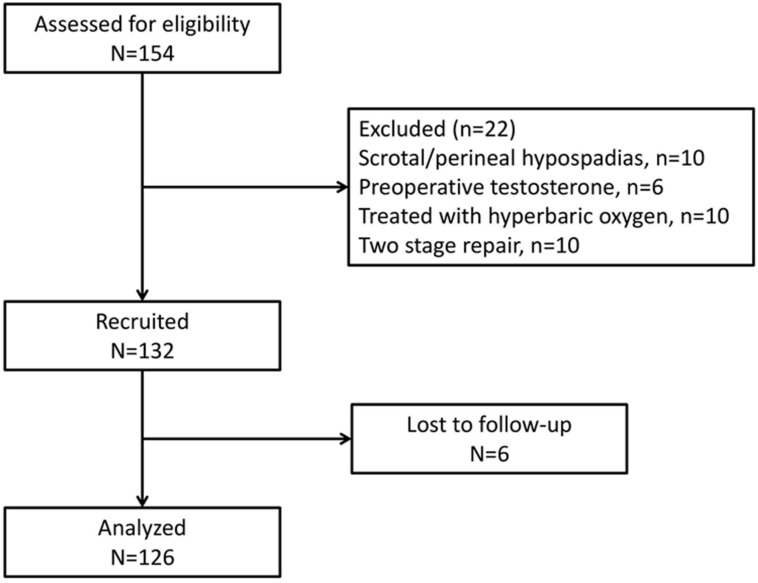
Patient flow diagram.

**Figure 2 children-11-00189-f002:**
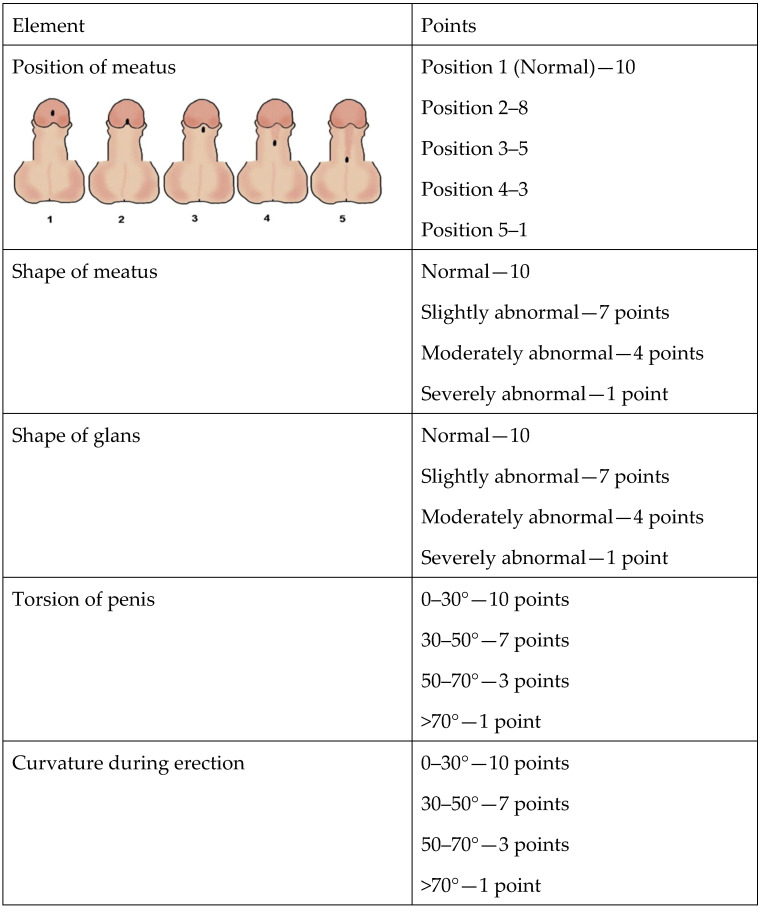
Elements of the Hypospadias Objective Penile Evaluation (HOPE) score.

**Figure 3 children-11-00189-f003:**
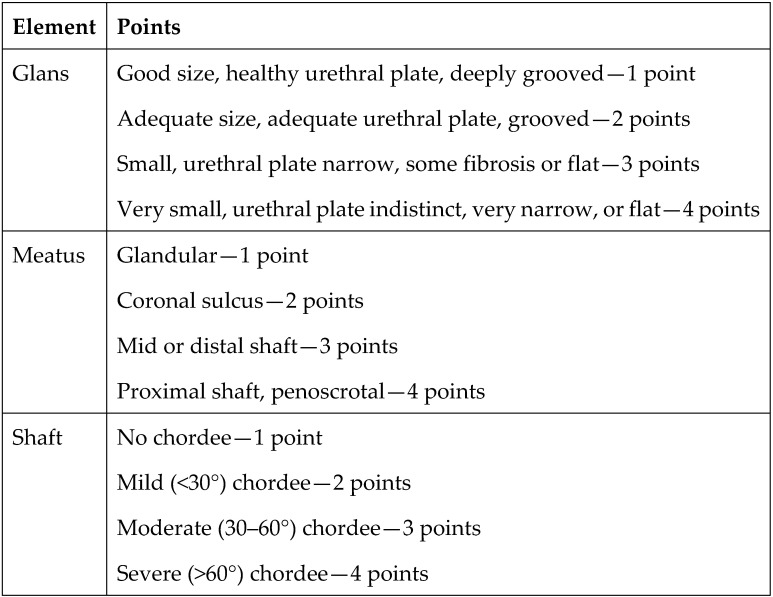
Elements of the Glans–Urethral Meatus–Shaft (GMS) score.

**Figure 4 children-11-00189-f004:**
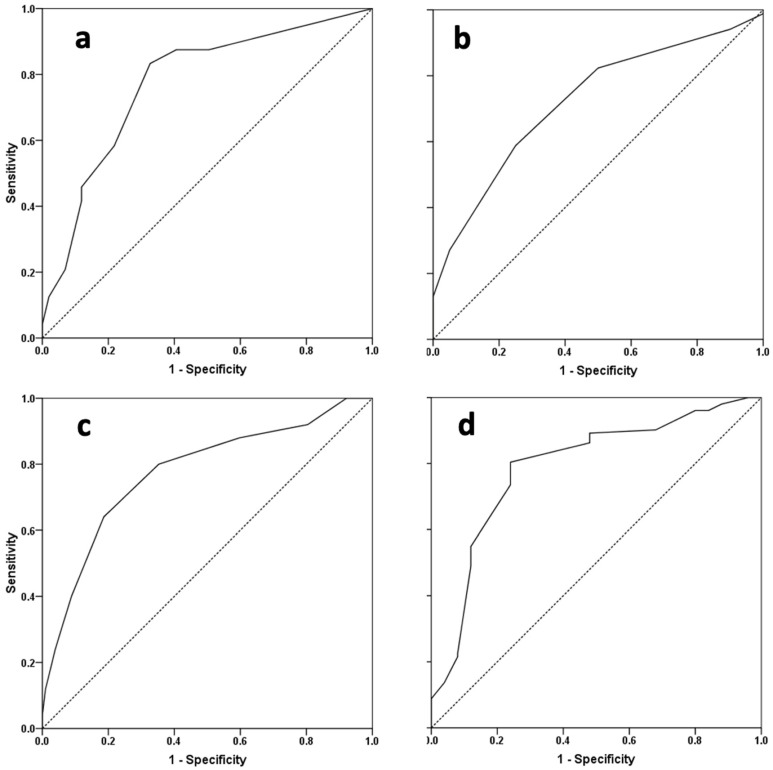
Receiver operating characteristic curves of (**a**) baseline chordee degree, (**b**) glans width, (**c**) GMS score, and (**d**) preoperative HOPE score as markers of optimal cosmetic outcomes (HOPE score ≥ 57).

**Table 1 children-11-00189-t001:** Comparison of demographics, clinical, and surgical parameters between boys with optimal (≥57) and suboptimal (<57) HOPE scores.

Variable	HOPE ≥ 57	HOPE < 57	*p*-Value
	*n* = 102 (81%)	*n* = 24 (19%)	
Birth weight (g)	3125 ± 613	3125 ± 716	0.99
Week of birth	38 ± 2.2	38 ± 1.8	0.63
Circumcised (%)	38 (37)	10 (40)	0.94
Meatal location (%)			0.013
Subcoronal	74 (73)	13 (54)	
Shaft	26 (25)	6 (25)	
Proximal	2 (2)	5 (21)	
Chordee (%)			<0.001
Mild	68 (67)	5 (20)	
Moderate	28 (27)	14(60)	
Severe	6 (6)	5 (20)	
Degree of Chordee (°)	10 (0–30)	40 (30–50)	<0.001
Glans width (mm)	13.8 ± 1.5	12.7 ± 1.1	0.003
Preoperative HOPE score	47 ± 4.8	41 ± 5	<0.001
GMS score	6 ± 1.8	8 ± 2.1	<0.001
GMS score severity, *n* (%)			<0.001
1	66 (65)	5 (21)	
2	32 (31)	13 (54)	
3	4 (4)	6 (25)	
Penile straightening procedure, *n* (%)	50 (51)	17 (71)	<0.001
Boys requiring ancillary procedure secondary to UC, *n* (%)	10 (10)	11 (46)	<0.001
Overall number of operations	1 (1–2)	2 (1–3)	<0.001
Days with stent	7.3 ± 2.1	7.6 ± 2.3	0.52

HOPE—Hypospadias Objective Penile Evaluation; GMS—Glans–Urethral Meatus–Shaft; UC—Urethroplasty complications.

**Table 2 children-11-00189-t002:** ROC curve analysis of potential predictors of optimal cosmetic outcomes (HOPE score ≥ 57).

Variable	AUC	OR (95% CI)	*p*-Value	Cutoff	Sensitivity (%)	Specificity (%)
Chordee degree	0.77	0.67–0.87	<0.001	25°	83	67
Glans width	0.72	0.60–0.84	0.002	12.5 mm	82	50
GMS score	0.78	0.67–0.88	<0.001	7	80	65
Preoperative HOPE score	0.79	0.69–0.89	<0.001	45	76	76

ROC—Receiver operating characteristic; AUC—Area under the curve; OR—Odds ratio; CI—Confidence interval; GMS—Glans–Urethral Meatus–Shaft; HOPE—Hypospadias Objective Penile Evaluation.

**Table 3 children-11-00189-t003:** Suboptimal postoperative HOPE score (<57) predictability according to univariate and multivariate logistic regression analyses.

Variable	Univariate		Multivariate *	
	OR (95% CI)	*p*-Value	OR (95% CI)	*p*-Value
Chordee degree	1.05 (1.02–1.07)	<0.001	1.04 (1.01–1.07)	0.013
Chordee degree ≥ 25°	10.3 (3.26–32.6)	<0.001	7.30 (1.74–30.56)	0.007
Glans width	0.55 (0.37–0.83)	0.005	0.52 (0.34–0.80)	0.003
Glans width ≥ 12.5 mm	0.21 (0.08–0.61)	0.005	0.20 (0.07–0.63)	0.005
Meatal location				
Subcoronal	Ref.	-	Ref.	-
Shaft	1.53 (0.55–4.26)	0.4	1.18 (0.38–3.63)	0.77
Proximal	14.2 (2.5–81.2)	0.005	7.54 (0.43–130.86)	0.17
GMS score	1.75 (1.24–2.29)	<0.001	1.66 (1.15–2.38)	0.007
GMS score ≥ 7	7.3 (2.53–21.1)	<0.001	4.31 (1.31–14.23)	0.017
Preoperative HOPE score	0.80 (0.73–0.89)	<0.001	0.82 (0.73–0.93)	0.002
Preoperative HOPE score ≥ 45	0.1 (0.04–0.28)	<0.001	0.15 (0.05–0.49)	0.002

All variables presented in this table were separately modeled as independent variables, whereas a urethroplasty complication was a dependent variable in all models. * Multivariable models were adjusted for birth weight, gestational age at birth, age at surgery, family history of hypospadias, neonatal circumcision, and type of urethroplasty. HOPE—Hypospadias Objective Penile Evaluation; OR—Odds ratio; CI—Confidence interval; GMS—Glans–Urethral Meatus–Shaft.

## Data Availability

Data will be made available upon request. Data is not available due to privacy reasons.

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
