# Peer review of "Use of Validated Questionnaires to Predict Cosmetic Outcomes of Hypospadias Repair"

_children, 2024, doi:10.3390/children11020189_

Round 1

Reviewer 1 Report

Comments and Suggestions for Authors

In their prospective cohort study, the authors identified predictors of cosmetic outcome after hypospadias repair and evaluated the reoperation rate due to cosmetic considerations in 126 male patients who had undergone primary hypospadias repair using the Glans-Urethral Meatus-Shaft (GMS) hypospadias score and the Hypospadias Objective Penile Evaluation (HOPE) score.

I read the study with great interest. Significant problems with the study design and methodology were identified. My concerns are as follows:

1. Abstract – The results in the abstract are inadequately presented. The authors should present the results of the main variables together with the p-values instead of general statements.

2. Since this is a retrospective study, I wonder why patients after 2020 were not included in this study. Including patients from later years would significantly improve the sample size and the quality of the study. In addition, the exact study period (including dates and months) should be specified.

3. The main problem with this study is the lack of sample size calculation. Since this is a prospective study, sample size calculation is mandatory. I do not see that the authors have done a sample size calculation. How do the authors know that the present sample size is sufficient to achieve statistical significance!

4. The next important issue is the high risk of bias in the evaluation of treatment outcomes. As the authors state, only one surgeon treated and evaluated all patients. At least two independent surgeons should be involved in the evaluation of patients, and the median score of two objective assessors should be used.

5. The authors analyzed three different types of hypospadias (subcoronal, shaft and proximal). This could also influence the results, as better results can be expected with more distal hypospadias. This objection is supported by the fact that the authors reported significant differences in their results between the groups in terms of hypospadias severity, glans width, preoperative HOPE score and preoperative GMS score. This is not an optimal strategy, and in my opinion this distribution is not acceptable in the analysis.

6. Why were the results of a single surgeon analyzed? Why were other surgeons not included?

7. Statistical analysis – it is unclear which statistical test was used to test the normality of the data distribution.

8. The resolution of Figure 1 is of very low quality. It should be replaced by an image of better quality. In addition, Figure 1 and the associated text should be replaced in the methodology. It is not a result.

9. Since the authors performed a ROC analysis, the ROC curve should be included as a separate figure.

10. The reference list should be updated. According to MDPI standards, at least 30 references are required for this type of study.

11. The quality of the English language needs to be improved. The manuscript should be edited by a native English speaker or a professional language editor to improve grammar and readability.

Comments on the Quality of English Language

The quality of the English language needs to be improved. The manuscript should be edited by a native English speaker or a professional language editor to improve grammar and readability.

Author Response

In their prospective cohort study, the authors identified predictors of cosmetic outcome after hypospadias repair and evaluated the reoperation rate due to cosmetic considerations in 126 male patients who had undergone primary hypospadias repair using the Glans-Urethral Meatus-Shaft (GMS) hypospadias score and the Hypospadias Objective Penile Evaluation (HOPE) score.

Response: Thank you for the insightful review. Please see our responses below.

I read the study with great interest. Significant problems with the study design and methodology were identified. My concerns are as follows:

  1. Abstract – The results in the abstract are inadequately presented. The authors should present the results of the main variables together with the p-values instead of general statements.

Response: We revised the abstract and added the results of the main variables with the p-values.

  1. Since this is a retrospective study, I wonder why patients after 2020 were not included in this study. Including patients from later years would significantly improve the sample size and the quality of the study. In addition, the exact study period (including dates and months) should be specified.

Response: Patients were recruited prospectively after obtaining institutional Ethics Committee approval and parents’ informed consent. We perform approximately 100 cases of hypospadias repair per year; of which, about 40 are primary cases that were eligible to participate in the study.

We based our sample size calculation (see details below) on an assumption of a 15% dropout rate, which indicated that we needed at least 22 patients in each study arm (optimal and sub-optimal cosmetic outcomes). The actual dropout rate was 5%. After recruiting 126 cases, the smaller group (suboptimal cosmetic outcomes) included 24 cases, and the recruitment was terminated.

We added the study period to the manuscript, page 8. ‘”The study was performed from January 2016 to February 2020’’.

  1. The main problem with this study is the lack of sample size calculation. Since this is a prospective study, sample size calculation is mandatory. I do not see that the authors have done a sample size calculation. How do the authors know that the present sample size is sufficient to achieve statistical significance!

Response:  We added details on the sample size calculation to the Data analysis section, page 11: “Based on the sample size calculation, we planned to include 136 patients in the study. The sample size was calculated using G∗Power software, version 3.1.9.7. The calculated sample size (at least 22 patients in each group) allowed us to have an 80% chance with a two-sided significance of 0.05, to detect a difference of 3 points in preoperative HOPE scores between boys with optimal (≥57) and sub-optimal (<57) HOPE scores. A difference of 3 points in preoperative HOPE score may be clinically significant in predicting optimal cosmetic outcomes. In the sample size calculation, we also considered an incidence of undesirable cosmetic results of the surgery of about 20% and a potential dropout rate of approximately 15%.“

  1. The next important issue is the high risk of bias in the evaluation of treatment outcomes. As the authors state, only one surgeon treated and evaluated all patients. At least two independent surgeons should be involved in the evaluation of patients, and the median score of two objective assessors should be used.

Response: We agree with the reviewer that an evaluation by several independent surgeons can potentially influence the results. We explained this on page 23. However, in the HOPE score validation study, the intra-observer reliability had an average Pearson correlation coefficient of 0.817, reflecting strong intra-rater reproducibility. The inter-observer reliability demonstrated an intraclass correlation coefficient of 0.790, indicating sufficient agreement among observers. These results imply that additional evaluation are unlikely to change the results dramatically[7]  (van der Toorn, F.; de Jong, T.P.V.M.; de Gier, R.P.E.; Callewaert, P.R.H.; van der Horst, E.H.J.R.; Steffens, M.G.; Hoebeke, P.; Nijman, R.J.M.; Bush, N.C.; Wolffenbuttel, K.P.; et al. Introducing the HOPE (Hypospadias Objective Penile Evaluation) score: a validation study of an objective scoring system for evaluating cosmetic appearance in hypospadias patients. J Pediatr Urol 2013, 9, 1006–1016, doi:10.1016/j.jpurol.2013.01.015.). Child-parent-doctor relationships in a research setup should be well-controlled. Page 23:  “All boys and parents meet with the same nurse practitioner and treating physician. Other personnel do not enter when the genitalia of an individual are being examined, for child-parent privacy and medico-legal reasons. It would be considered inappropriate in our institution to have more than one doctor examine the child, especially when the genitalia are being examined [31].” (Luchtenberg ML, Maeckelberghe ELM, Locock L, Verhagen AAE. Understanding the child-doctor relationship in research participation: a qualitative study. BMC Pediatr. 2020 Jul 24;20(1)).

  1. The authors analyzed three different types of hypospadias (subcoronal, shaft and proximal). This could also influence the results, as better results can be expected with more distal hypospadias. This objection is supported by the fact that the authors reported significant differences in their results between the groups in terms of hypospadias severity, glans width, preoperative HOPE score and preoperative GMS score. This is not an optimal strategy, and in my opinion this distribution is not acceptable in the analysis.

Response: We thank the reviewer for the comment. Subcoronal, distal shaft, midshaft, and proximal hypospadias are all located along the penile shaft and treated with the same one-stage surgical techniques. Therefore, we analyzed them together. Our results show that meatal location is not a predictor for cosmetic outcomes, which strengthens our decision to include all shaft hypospadias in the analysis. We excluded more proximal hypospadias (penoscrotal, scrotal, and perineal), as shown in the patient flow diagram because they require a staged surgical approach.

  1. Why were the results of a single surgeon analyzed? Why were other surgeons not included?

Response: Our medical group includes only one experienced, fully-trained surgeon who completed an American Urology Association (AUA) formal pediatric urology fellowship accredited by the Society of Pediatric Urology (SPU). In the limitations paragraph of the manuscript, the following sentence is included: (page 19-20) ‘”We present the experience of a single surgeon at one academic center. Therefore, the results may not be generalizable and should be confirmed by larger multi-center studies. However, surgery performed by a single surgeon removes the bias associated with variations in technique and decision-making processes associated with cohorts involving several surgeons. Outcomes were also evaluated by the same single surgeon.”

  1. Statistical analysis – it is unclear which statistical test was used to test the normality of the data distribution.

Response: The Kolmogorov-Smirnov test was conducted to determine whether the variables involved in our study were normally distributed. We added this information to the Statistics section (page 14): “The Kolmogorov-Smirnov test was conducted to determine whether the study variables were  normally distributed.”.

  1. The resolution of Figure 1 is of very low quality. It should be replaced by an image of better quality. In addition, Figure 1 and the associated text should be replaced in the methodology. It is not a result.

Response: We replaced the image for better resolution and moved it to the Materials and Methods section.

  1. Since the authors performed a ROC analysis, the ROC curve should be included as a separate figure  

Response: We added the ROC curve as Figure 4.

  1. The reference list should be updated. According to MDPI standards, at least 30 references are required for this type of study.

 Response: We thank the reviewer for this valid comment, after revisions the article now meets the MDPI standards and includes 34 references.

  1. The quality of the English language needs to be improved. The manuscript should be edited by a native English speaker or a professional language editor to improve grammar and readability.

Response: The manuscript was edited by a professional language editor. We hope it is more readable after this improvement.

Comments on the Quality of English Language

The quality of the English language needs to be improved. The manuscript should be edited by a native English speaker or a professional language editor to improve grammar and readability.

Response: We edited the revised manuscript with the help of native English-speaking scientific editor.

Reviewer 2 Report

Comments and Suggestions for Authors

In this paper, the authors aimed to identify preoperative predictors of cosmetic outcomes and to assess the reoperation rate according to cosmetic considerations. 

I read the article with great interest, and my suggestions for improvements are as follows.

1) In the second paragraph of the introduction, authors state: "[...] Typical reasons include fistula formation (5.7%), dehiscence, meatal stenosis (3.6%), and development of urethral stricture (1.3%)". if available, please add the missing percentage for "dehiscence".

2)In the materials and methods, section 2.1 - second paragraph, the authos should better clarify why they assessed the degree of curvature after degloving instead of before.

3)In the materials and methods, section 2.2 - first paragraph, authors should better specify which type of stent they used (foley catheter, nelaton catheter, etc.) and whether they used an 8 Fr catheter for all patients.

4) The quality of Figure 1 should be improved if possible.

5) Please, combine the limitations of the study in a single paragraph.

6) The conclusions are poor and must be improved based on the results of the study.

Author Response

In this paper, the authors aimed to identify preoperative predictors of cosmetic outcomes and to assess the reoperation rate according to cosmetic considerations.

Thank you for the insightful review. We carefully addressed each comment. Please see our responses below.

I read the article with great interest, and my suggestions for improvements are as follows.

  1. In the second paragraph of the introduction, authors state: "[...] Typical reasons include fistula formation (5.7%), dehiscence, meatal stenosis (3.6%), and development of urethral stricture (1.3%)". if available, please add the missing percentage for "dehiscence".

Response: We added the percentage of dehiscence to the Introduction (2-15%) and reference 4 (Snodgrass W, Cost N, Nakonezny PA, Bush N. Analysis of risk factors for glans dehiscence after tubularized incised plate hypospadias repair. J Urol 2011;185, 1845-1849).

  1. In the materials and methods, section 2.1 - second paragraph, the authors should better clarify why they assessed the degree of curvature after degloving instead of before.

Response: We added the following paragraph and reference to the Materials and Methods section: “Correction of chordee is dictated by its severity , as described in a meta-analysis by Babu et al. [11].” (Babu, R.; Chandrasekharam, V.V.S. A meta-analysis comparing dorsal plication and ventral lengthening for chordee correction during primary proximal hypospadias repair. Pediatr Surg Int 2022, 38, 389–398, doi:10.1007/s00383-022-05065-7.). In the 17 articles included in the meta-analysis, the severity of chordee was evaluated after degloving [12]. (Moscardi, P.R.M.; Gosalbez, R.; Castellan, M.A. Management of High-Grade Penile Curvature Associated With Hypospadias in Children. Front Pediatr 2017, 5, 189, doi:10.3389/fped.2017.00189). The  surgical technique was chosen after degloving and releasing tethering bands [13].” (Keays, M.A.; Dave, S. Current hypospadias management: Diagnosis, surgical management, and long-term patient-centred outcomes. CUAJ 2017, 11, 48, doi:10.5489/cuaj.4386.)

  1. In the materials and methods, section 2.2 - first paragraph, authors should better specify which type of stent they used (foley catheter, nelaton catheter, etc.) and whether they used an 8 Fr catheter for all patients.

Response: We added the following sentence to the description in the Surgical technique paragraph in  the Materials and Methods section: “Urethroplasty was performed over an 8F stent. We used an 8F Zaontz stent (Cook Medical, Bloomington, IN, USA) for all patients [15].”

  1. The quality of Figure 1 should be improved if possible.

              Response: The quality of Figure 1 was improved, although the article has been reorganized and it is now Figure 3.

  1. Please, combine the limitations of the study in a single paragraph.

Response: We added a separate heading for Limitations and revised the section for readability and clarity, as other reviewers requested additional limitations to the study. We made an effort to restrict the limitation paragraph as much as possible.

  1. The conclusions are poor and must be improved based on the results of the study.

Response: We expanded on the Conclusions based on the results of the study, adding the following: “More than 10% of cases undergo ancillary procedures secondary solely to cosmetic considerations. There are multiple predictors of optimal cosmetic outcomes after hypospadias surgery: degree of chordee, glans width, and preoperative HOPE and GMS scores, which are the best predictors of satisfactory cosmetic results. Although meatal location is the main predictor for postoperative complications, it is not a predictor for cosmetic outcomes. Parents should acknowledge the factors affecting cosmetic outcomes during preoperative consultation. Factors affecting cosmetic outcomes should be clearly explained to parents during the preoperative consultation.”

Reviewer 3 Report

Comments and Suggestions for Authors

The manuscript is well structured. However, I have several questions.

It is essential to explain the items that assess each of the two scales in the paper, both the GMS and the HOPE. A figure or a table explaining the items assessed by each of them should be added to facilitate readers' understanding of the manuscript.

I miss the ROC curve in a Figure, the data provided in table 2.

Regarding Table 2, most of the variables are continuous, and therefore can be expressed and compared in a ROC curve. However, the location of the urethral meatus is a qualitative variable, and therefore cannot be compared in this type of analysis.

The main pitfall of this work is that the HOPE scale measures postoperative aesthetic outcomes. It is difficult to justify the preoperative scoring of this scale. The GMS scale, chorda grades or glans width are also associated with a worse cosmetic outcome in the multivariate analysis. However, only the HOPE and GMS scales are referred to in the conclusions. 

The bias of preoperatively measuring a scale that assesses postoperative outcome is an important limitation in this manuscript.

Comments on the Quality of English Language

The manuscript is well structured. However, I have several questions.

It is essential to explain the items that assess each of the two scales in the paper, both the GMS and the HOPE. A figure or a table explaining the items assessed by each of them should be added to facilitate readers' understanding of the manuscript.

I miss the ROC curve in a Figure, the data provided in table 2.

Regarding Table 2, most of the variables are continuous, and therefore can be expressed and compared in a ROC curve. However, the location of the urethral meatus is a qualitative variable, and therefore cannot be compared in this type of analysis.

The main pitfall of this work is that the HOPE scale measures postoperative aesthetic outcomes. It is difficult to justify the preoperative scoring of this scale. The GMS scale, chorda grades or glans width are also associated with a worse cosmetic outcome in the multivariate analysis. However, only the HOPE and GMS scales are referred to in the conclusions. 

The bias of preoperatively measuring a scale that assesses postoperative outcome is an important limitation in this manuscript.

Author Response

The manuscript is well structured. However, I have several questions.

Thank you for the insightful review. We carefully addressed each comment. Please see our responses below.

  1. It is essential to explain the items that assess each of the two scales in the paper, both the GMS and the HOPE. A figure or a table explaining the items assessed by each of them should be added to facilitate readers' understanding of the manuscript.

Response: Thank you for this comment. We added a section titled Study tools and Figures (Figures 2 and 3) to provide additional details on the HOPE and GMS scores. We believe this will help clarify the items for the reader.

  1. I miss the ROC curve in a Figure, the data provided in table 2.

Response: We added the ROC curves to the manuscript. Please see Figure 4. “Figure 4: Receiver operating characteristic curves of: (a) baseline chordee degree, (b) glans width, (c) GMS score, and (d) preoperative HOPE score as markers of optimal cosmetic outcomes (HOPE score ≥57).”

  1. Regarding Table 2, most of the variables are continuous, and therefore can be expressed and compared in a ROC curve. However, the location of the urethral meatus is a qualitative variable, and therefore cannot be compared in this type of analysis.

Response: We thank the reviewer for this comment. We removed the variable from the table.

  1. The main pitfall of this work is that the HOPE scale measures postoperative aesthetic outcomes. It is difficult to justify the preoperative scoring of this scale. The GMS scale, chorda grades or glans width are also associated with a worse cosmetic outcome in the multivariate analysis. However, only the HOPE and GMS scales are referred to in the conclusions.

Response: We thank the reviewer for these two comments.

Regarding the preoperative use of the HOPE score, the aim of the study was to define factors that can predict cosmetic outcomes, as this is the first study on the subject and risk factors were not yet defined, we included known risk factors for urethroplasty complications in the univariate and multivariable models: meatal location, chordee, glans width and GMS score. The HOPE score is a tool to assess cosmetic appearance. We used it as one of the variables in the multivariable model, and indeed, it proved to be a predictor. Although the reviewer stated this is the main pitfall of the study, we would like to emphasize that it is only one of the parameters that we explored. Furthermore, in other fields of surgery where cosmetic outcomes are an issue, such as rhinoplasty and breast surgery, the same questionnaires were used pre- and postoperatively. Patniak et al. used the Derriford Appearance Scale for the evaluation of cosmetic outcomes after rhinoplasty. The same scale was used preoperatively and 3 months postoperatively. Rosa et al. validated the Utrecht questionnaire for outcome assessment in aesthetic rhinoplasty and used the same questionnaire pre- and postoperatively. Ghilli et al. reported quality of life and cosmetic satisfaction using the same BREAST-Q questionnaire pre-and post-operatively.

We added the following sentence to the Limitations paragraph: “Additionally, using the HOPE score for both preoperative assessment and as an outcome measure has the potential to introduce bias. Given the absence of currently validated tools to assess preoperative cosmetic appearance, we utilized the HOPE score for this purpose, like other surgical procedures where cosmetic outcomes are crucial. In the literature, Patniak et al. employed the Derriford Appearance Scale to evaluate cosmetic outcomes in rhinoplasty. The same scale was used both preoperatively and three months postoperatively[32]. Rosa et al. validated the Utrecht questionnaire for outcome assessment in aesthetic rhinoplasty; the same questionnaire was used pre and postoperatively [33] Ghilli et al. reported on the quality of life and cosmetic satisfaction using the BREAST-Q questionnaire before and after oncoplastic or traditional breast-conserving surgery [34]”.

New references:

  1. Patnaik, U.; Nilakantan, A.; Bajpai, R.; Addya, K. Comprehensive assessment in cosmetic rhinoplasty: The use of the Derriford Appearance Scale for evaluation of patients. Med J Armed Forces India 2019, 75, 184–189, doi:10.1016/j.mjafi.2018.07.011.
  2. Rosa, F.; Lohuis, P.J.F.M.; Almeida, J.; Santos, M.; Oliveira, J.; Sousa, C.A.E.; Ferreira, M. The Portuguese version of “The Utrecht questionnaire for outcome assessment in aesthetic rhinoplasty”: validation and clinical application. Braz J Otorhinolaryngol 2019, 85, 170–175, doi:10.1016/j.bjorl.2017.11.007.
  3. Ghilli, M.; Mariniello, M.D.; Ferrè, F.; Morganti, R.; Perre, E.; Novaro, R.; Colizzi, L.; Camilleri, V.; Baldetti, G.; Rossetti, E.; et al. Quality of life and satisfaction of patients after oncoplastic or traditional breast-conserving surgery using the BREAST-Q (BCT module): a prospective study. Breast Cancer 2023, 30, 802–809, doi:10.1007/s12282-023-01474-1.

Response 2: Regarding the reviewer’s second comment, we added the chordee grade and glans width as predictors of satisfactory cosmetic outcomes to the Conclusions. “Predictors of optimal cosmetic outcomes after hypospadias surgery include degree of chordee, glans width, and preoperative HOPE and GMS scores, which are the best predictors of satisfactory cosmetic results.”

  1. The bias of preoperatively measuring a scale that assesses postoperative outcome is an important limitation in this manuscript.

Response: We acknowledge this bias as a potential limitation of the study. We added the following information to the Limitations: “Additionally, using the HOPE score for both preoperative assessment and as an outcome measure has the potential to introduce bias. Given the current absence of validated tools to assess preoperative cosmetic appearance, we used the HOPE score for this purpose, like other surgical procedures where cosmetic outcomes are crucial. Patniak et al. employed the Derriford Appearance Scale to evaluate cosmetic outcomes in rhinoplasty. The same scale was used both preoperatively and three months postoperatively [32]. Rosa et al. validated the Utrecht questionnaire for outcome assessment in aesthetic rhinoplasty, using the same questionnaire pre- and postoperatively [33]. Ghilli et al. reported on quality of life and cosmetic satisfaction using the BREAST-Q questionnaire before and after oncoplastic or traditional breast-conserving surgery [34]”.

Round 2

Reviewer 1 Report

Comments and Suggestions for Authors

The manuscript was significantly improved after revision, especially in terms of methodology and study design. In my opinion, the manuscript is acceptable in its present form.

Comments on the Quality of English Language

--

Reviewer 3 Report

Comments and Suggestions for Authors

The authors have correctly adapted the manuscript to the reviewers' suggestions. Publication can be considered in the current state of the article.

Comments on the Quality of English Language

Minor editing of English language required